# Occupational Burnout in Nurses and Corporate Employees in Małopolska Region, Poland

**DOI:** 10.3390/healthcare13020123

**Published:** 2025-01-09

**Authors:** Anna Nowacka, Agnieszka Gniadek, Agnieszka Micek, Paulina Świątek, Tadeusz Wadas, Renata Wolfshaut-Wolak

**Affiliations:** 1Department of Nursing Management and Epidemiological Nursing, Faculty of Health Sciences, Jagiellonian University Medical College, 31-008 Krakow, Poland; agnieszka.gniadek@uj.edu.pl; 2Statistics Laboratory, Faculty of Health Sciences, Jagiellonian University Medical College, 31-008 Krakow, Poland; agnieszka.micek@uj.edu.pl; 3Institute of Nursing and Midwifery, Faculty of Health Sciences, Jagiellonian University Medical College, 31-008 Krakow, Poland; ppaulinaswiatek@gmail.com; 4Małopolska District Chamber of Nurses and Midwives in Kraków, 31-153 Krakow, Poland; twadas@moipip.org.pl

**Keywords:** occupational burnout, nurses, corporate employees

## Abstract

**Introduction**: Work is an essential aspect of human life. However, high expectations from employers and clients, combined with time pressure and chronic stress, can contribute to burnout among employees in service professions. **Aim of the Study**: This study aimed to compare the prevalence of burnout syndrome between two occupational groups—corporate office workers and active nurses—and to assess the influence of socio-demographic factors on the level of burnout in both groups. **Materials and Methods**: The study was conducted among 330 participants, including 126 nurses (38%) and 204 corporate employees (62%). Data were collected using a standard questionnaire and the Maslach Burnout Inventory. The study period spanned from November 2018 to October 2019, and questionnaires were distributed via email. **Results**: The findings revealed that both corporate employees and nurses experience occupational burnout. Both groups predominantly exhibited moderate to high levels of emotional exhaustion (nurses: 66.67%, *q*2 = 20.5; corporate employees: 73.53%, *q*2 = 31.5) and low to moderate levels of personal accomplishment (nurses: 78.57%, *q*2 = 32.0; corporate employees: 87.75%, *q*2 = 27.0). Corporate employees showed significantly higher depersonalization scores (*q*2 = 13.50) compared to nurses (*q*2 = 5.0). The prevalence of burnout in both groups was influenced by socio-demographic factors, including having children, their place of residence, education, earnings, and job-related factors such as vacation availability, the frequency of work shifts, work systems, overtime, and overall job satisfaction. **Conclusions**: Service-oriented professions that involve caregiving and direct interpersonal interactions face comparable risks of occupational burnout. Despite differences in the nature and conditions of work, as well as tasks performed, both corporate employees and active nurses are vulnerable to burnout. High salaries were not confirmed as a protective factor against occupational burnout. Additionally, environmental factors, particularly those related to working conditions, played a significant role in the development of burnout syndrome, alongside individual factors.

## 1. Introduction

An organization where a person is employed can either accelerate their burnout or create healthy and favorable conditions for their well-being. For years, Poles have been among the longest and hardest-working nationalities. A study by the European Foundation for the Improvement of Living and Working Conditions revealed that in 2020, the average Pole worked 1848 h per year—the highest figure in the European Union. In comparison, employees in Germany worked 1574 h, and those in France worked 1610 [1]. However, data from Eurostat for 2021 indicates that “sitting around” at work does not necessarily translate into economic growth for Poland. Longer working hours do not equate to higher productivity. For example, the average Pole earns USD 42 per hour, compared to USD 128.2 for workers in Ireland, USD 99 in Luxembourg, and USD 84.4 in Norway [2]. This raises the question: what accounts for the low productivity of Polish workers?

One contributing factor is occupational stress, which arises when work becomes excessively physically or mentally taxing. A report, The Workforce View in Europe 2017, noted that 22% of Polish workers experience daily stress, and 24% experience it weekly [3]. Similarly, the report Stress at Work and Its Impact on the Occurrence of Occupational Accidents and the Health of Working People, published by the Healthy Work Association and commissioned by the Social Insurance Institution (ZUS), estimated that in 2013, stress-related absences amounted to 77,888.33 lost workdays, costing approximately PLN 9.84 billion per year [4]. Chronic workplace stress can lead to occupational burnout, which is a growing issue in today’s labor market that can be apply to every employee. According to the International Classification of Diseases and Health Problems (ICD-11), burnout is defined as a syndrome resulting from chronic workplace stress that is not effectively managed [5,6].

Christina Maslach conceptualized burnout as encompassing three dimensions: emotional exhaustion, which includes feelings of emptiness, a lack of strength, and energy depletion; depersonalization, characterized by a cynical outlook on others, reduced sensitivity, callousness, and a lack of interest or concern for others; and a lack of personal accomplishment, which involves a diminished evaluation of one’s achievements, feelings of ineffectiveness, and sense incompetence [7,8,9].

Several factors contribute to burnout. Workplace-related factors include improper work organization, excessive workload, monotonous tasks, conflicting requirements, lack of control or decision-making power, and inadequate remuneration for effort. Interpersonal factors involve a lack of mutual support or trust, hostile rivalry, conflicts, poor interpersonal communication, unfair treatment, and workplace bullying (mobbing). Employee-related factors, such as perfectionism, excessive work commitment, over-identification with the company, a strong need for appreciation, and poor self-organization skills, also play a role. It is challenging to determine which of these factors has the greatest impact on the occurrence of burnout [10,11,12,13].

In Poland, there is a growing interest in the phenomenon of professional burnout, particularly in social service professions such as nursing, medicine, teaching, and social work. These roles involve helping others and maintaining close interpersonal contact. Burnout is also prevalent among employees of large companies and corporations, where work is performed in a multitasking mode under immense pressure related to responsibility and time. These employees often work long hours, exceeding 8 h per day. Corporations are frequently perceived negatively as they impose constant control, monitor work performance, and enforce strict procedures. These practices significantly restrict employees’ ability to use their competencies and skills, leading to feelings of irrelevance and a lack of control over their work. This sense of disempowerment can extend to a perceived lack of control over one’s life, which may directly contribute to an increased risk of chronic diseases, including life-threatening conditions.

Excessive workloads leave little time for establishing meaningful relationships. While the development of information technology has improved long-distance communication and enhanced work efficiency, it has also depersonalized human interactions [14]. Fierce competition in the corporate world can sometimes pressure employees to act against their values. A common example is call centers or telemarketing roles, where employees may feel compelled to persuade customers to purchase products that do not align with the descriptions provided, effectively engaging in deception. Such roles, which are often low paid, can quickly lead to frustration, discouragement, and burnout. In addition, bureaucratic restrictions in these environments significantly inhibit the employees’ freedom [1,8,15,16].

Unlike professions such as medicine or nursing, office work is considered light, requiring no physical exertion, and typically performed in a sitting position in front of a computer. However, prolonged sedentary work in a constrained posture, combined with visual strain, limited musculoskeletal mobility, and poor dietary habits, can lead to the development of various health issues. These include diseases of the cardiovascular, musculoskeletal, and nervous systems. Office workers frequently report symptoms such as eyestrain, impaired vision, weight gain, and a general decline in physical health and performance [16].

Occupational burnout is undoubtedly a complex and multifaceted phenomenon, making it challenging to describe comprehensively. This study focuses on comparing the incidence of occupational burnout in two seemingly distinct groups: corporate employees and active nurses. These groups differ significantly in terms of job nature, responsibilities, and working conditions. The study also seeks to identify the factors contributing to the development of occupational burnout in these populations.

In Poland, corporate employees are often perceived as among the highest-paid workers, while medical professionals, including nurses, are believed to earn the least. There is a widespread assumption that low and unsatisfactory salaries are a primary cause of burnout across all sectors. This study aims, among other objectives, to verify this opinion. To the best of our knowledge, this work is one of the few studies attempting to challenge this perspective as similar research was not found in the existing literature.

## 2. Aim of the Study

The study aimed to compare the prevalence and intensity of burnout syndrome in two occupational groups associated with the social service profession: corporate office workers and active nurses.

## 3. Materials and Methods

A cross-sectional, descriptive, and correlational design was used to compare the prevalence of burnout syndrome in two occupational groups: nurses and corporate employees.

The study was conducted between December 2018 and June 2019 in the Malopolska region of Poland and had a multicenter character. Participation in the study was voluntary and anonymous, with respondents participating as volunteers. The inclusion criteria for the study were as follows: providing written consent to participate, employment as a nurse or corporate employee, and a workplace in either a hospital or corporation. The exclusion criteria were as follows: lack of consent to participate and lack of employment as a nurse or corporate employee.

A total of 420 individuals were invited to participate, of whom 330 (78.6%) provided consent and were included in the analysis.

The authors of the study were not employees of the corporations or medical institutions involved and did not receive any financial benefits from the research. The study received approval from the Bioethics Committee of the Jagiellonian University Collegium Medicum (Approval No. 1072.6120.284.2018, dated 25 October 2018) and consent from the principals of the participating medical entities and corporations.

The study utilized a questionnaire consisting of two parts. The first part gathered data on socio-demographic variables such as gender, age, education, place of residence, marital status, having children, household size, and employment history. Employment-related questions included length of service, work system, bonus system, working on weekends or public holidays, night shifts, additional work, frequency of job changes, gross monthly income per person in the household, self-assessment of financial situation, and whether the participant had taken a holiday in the previous year.

The second part of the study utilized the MBI-HSS (Maslach Burnout Inventory Human Services Survey) Professional Burnout Questionnaire, which examines three aspects of occupational burnout: emotional exhaustion (EE), depersonalization (DEP), and a reduced sense of personal accomplishment (PA). The questionnaire consists of 22 statements, each assigned to one of three separate subscales. The emotional exhaustion subscale includes nine statements (1, 2, 3, 6, 8, 13, 14, 16, and 20). The depersonalization subscale includes five statements (5, 10, 11, 15, and 22). The personal accomplishment subscale includes eight statements (4, 7, 9, 12, 17, 18, 19, and 21). Respondents rated the frequency with which they experienced the feelings or attitudes described in each statement on a seven-point scale, with options ranging from 0 to 6: 0—never, 1—several times a year, 2—once a month, 3—several times a month, 4—once a week, 5—several times a week, and 6—daily. The results for each subscale were analyzed by first determining the mean value for each subscale and then categorizing the scores into corresponding levels. The reference norms for each subscale were as follows: Emotional exhaustion: high (≥27), medium (17–26), and low (0–16). Depersonalization: high (≥13), medium (7–12), and low (0–6). Personal accomplishment: high (≥39), medium (32–38), and low (0–31). Occupational burnout was identified in participants with high scores on the emotional exhaustion and depersonalization subscales and low scores on the personal accomplishment subscale [17]. The internal consistency of the Polish adaptation of the MBI-HSS reached Cronbach’s α of 0.7 [18]. For this study, the original MBI questionnaire in Polish, purchased from the authors of the scale, was used.

### Statistical Analysis

The relationship between the subscales of occupational burnout and quantitative variables was examined using Spearman’s rank correlation. Since the scales analyzed in each category deviated from a normal distribution, the distribution of individual dimensions of occupational burnout across categories of socio-demographic characteristics and occupational variables was compared using either the Mann–Whitney *U* test (for variables with two categories) or the Kruskal–Wallis rank ANOVA test (for variables with three or more categories). The consistency of the psychosocial subscales was assessed using Cronbach’s alpha coefficient. Statistical analyses were conducted using the R statistical package (R version 3.5.0; Development Core Team, Vienna, Austria; http://www.r-project.org (accessed on 7 May 2024). A significance level of 0.05 was adopted.

## 4. Results

### 4.1. Demographic and Work Condition Variables

The study included 330 participants comprising active nurses employed in hospitals and corporate employees. Among the respondents, there were 126 nurses, 97.62% of whom were women, and 204 corporate employees, 30% of whom were men. The average age of the nurses was 37 years, with the youngest being 22 and the oldest 69. For corporate employees, the average age was 29 years, with ages ranging from 21 to 50. In both groups, the majority of respondents lived in urban areas (nurses: 66.67%; corporate employees: 95.10%) and held a university degree (nurses: 73.81%; corporate employees: 87.75%). Among nurses, 77.78% were in a relationship, compared to 66.67% of corporate employees. Additionally, 50% of nurses had children, compared to 21% of corporate employees. Most respondents, regardless of profession, lived in households consisting of 2–5 members (Table 1).

The average length of employment was similar in both groups. Nurses predominantly worked in a two-shift system, whereas corporate employees primarily worked in a single-shift system. Nurses’ work was not tied to a bonus system, unlike that of corporate employees. Nearly all nurses worked on weekends, public holidays, and night shifts, while over 60% of corporate employees reported similar work patterns. Approximately 25% of nurses and more than 14% of corporate employees reported working extra hours. Corporate employees change their workplace more frequently than nurses. Only 11% of corporate employees reported never having changed jobs, compared to 58% of nurses. Regarding financial status, 40% of nurses and 66% of corporate employees rated their financial situation as very good or good. Additionally, 50% of nurses and 77% of corporate employees reported taking a holiday in the past year (Table 2).

A total of 63% of corporate employees were satisfied with their work. Despite this, the desire to change the current job was expressed by as many as 59%. In the case of nurses, the vast majority of them—70%—expressed satisfaction with their job and did not want to change it—66%.

### 4.2. Prevalence of Occupational Burnout

Average and high levels of emotional exhaustion were observed in 66.67% of nurses and 73.53% of corporate employees surveyed. Nurses reported low levels on the depersonalization subscale in 65.9% of cases, whereas 79.96% of corporate employees reported medium and high levels on this subscale. On the personal accomplishment subscale, a low and medium sense of personal accomplishment was reported by the vast majority of nurses (78.57%) and corporate employees (87.75%). Detailed data are presented in Table 3.

### 4.3. Comparison of the Incidence of Burnout in the Study Groups

In both study groups, an attempt was made to examine the relationship between selected variables and the level of occupational burnout taking into account three dimensions: emotional exhaustion, depersonalization, and a reduced sense of personal accomplishment.

Table 4 presents differences in the incidence of occupational exhaustion among nurses and corporate employees in terms of socio-demographic and occupational factors. Statistically significant differences are in bold, and the values are given below the table.

After analysis, it was found that education significantly related the level of depersonalization in the group of nurses. Nurses with higher education displayed higher levels of depersonalization (5%) compared to nurses with secondary education (4%). However, no such relationship was observed among corporate employees.

Nurses with children exhibited higher levels of emotional exhaustion (23% vs. 18%) but lower levels of depersonalization (4% vs. 6%) compared to nurses without children. Among corporate employees, no such relationship was found, likely because the majority of respondents (79.41%) did not have children.

Corporate employees living in rural areas were more likely to have higher scores on the emotional exhaustion subscale (40.5% vs. 31%) and tended to have a higher sense of personal accomplishment compared to their urban counterparts (28% vs. 13.5%). In contrast, the place of residence did not significantly correlate with any of the subscales in the nurses’ group.

In corporate employees, emotional exhaustion was most prevalent among those working in a two-shift system (37%), while higher depersonalization scores were noted in two-shift and three-shift systems (19%). In addition, these individuals exhibited higher levels of personal accomplishment compared to single-shift employees, who reported the most reduced sense of personal accomplishment (28).

Among nurses, depersonalization was most frequently observed in those working three shifts (7.5%), while a reduced sense of personal accomplishment was most commonly reported among single-shift nurses (33%).

The frequency of job changes significantly associated the levels of depersonalization (never—5% vs. 1–3 times—4% vs. more than three times—3.5%). Similarly, reduced personal accomplishment was more severe among nurses who had never changed jobs (33%) compared to those who had changed jobs 1–3 times (32%) or more than three times (30%). Among corporate employees, the frequency of job changes did not significantly correlate with the level of occupational burnout.

Corporate employees who assessed their financial status as poor reported significantly higher levels of emotional exhaustion (40%) compared to those with average (37%) or good/very good financial status (27%). Interestingly, higher financial status correlated with greater job satisfaction but a lower sense of personal accomplishment (very good: 11%, average: 18%, and poor: 23%). Corporate employees with a poor financial situation also exhibited the highest levels of depersonalization and rated their accomplishments poorly, equating professional success with high salaries. Similar correlations between subjective financial assessments and burnout levels were observed among nurses, although these results were not statistically significant.

The study found that in both groups, respondents who had not taken holidays in the past year reported greater emotional exhaustion and lower job satisfaction. However, this variable did not have a statistically significant correlation on depersonalization values.

A relationship between job dissatisfaction and emotional exhaustion was observed in both groups. The higher the job dissatisfaction, the higher the emotional exhaustion values. Among nurses satisfied with their jobs, emotional exhaustion was less frequent (18% vs. 25%), and a similar trend was observed among corporate employees (22% vs. 42%). Additionally, corporate employees who were satisfied with their jobs reported a significantly lower frequency of reduced job satisfaction (10%) compared to dissatisfied employees (18%). The differences in emotional exhaustion between nurses and corporate employees were not statistically significant.

Similar trends were observed regarding willingness to change jobs. Respondents who expressed a desire to change jobs demonstrated higher levels of exhaustion—24% among nurses wanting to change jobs compared to 18% among those not wanting to change and 39% among corporate employees wanting to change compared to 33% among those intending to stay. This variable also influenced the sense of personal accomplishment: dissatisfied corporate employees who wanted to change jobs experienced reduced job satisfaction. In corporate employees, dissatisfaction with their jobs and the desire to change significantly depended on the depersonalization levels and reduced job satisfaction. Higher depersonalization values were observed in dissatisfied employees who wanted to change jobs (39%) compared to those who did not (19%). Similarly, reduced job satisfaction was more prevalent in employees intending to resign (17%) compared to those intending to stay (8.5%).

## 5. Discussion

The results of this study provide an assessment of the overall level of occupational burnout in both groups—corporate employees and nurses—across three subscales: emotional exhaustion, depersonalization, and personal accomplishment. They also allow for an evaluation of the relationship between selected variables and the incidence of professional burnout, as well as a comparison with findings from other studies. The results of our study indicate higher levels of occupational burnout among corporate employees than among nurses on all three subscales of occupational burnout. High or medium emotional exhaustion was present in 73.53% (*q*2 = 31.5) of corporate employees and 66.67% of nurses (*q*2 = 20.5). The level of depersonalization in the group of corporate employees reached higher values than in nurses 76.96% (*q*2 = 13.50) vs. 34.13 (*q*2 = 5). In turn, the sense of personal accomplishment characterizing the studied nurses turned out to be higher than in corporate employees 21.43% (*q*2 = 32.0) vs. 12.25% (*q* = 27.0). The average and high scores obtained on the subscale of emotional exhaustion may have resulted from physical and mental strain at work manifested by fatigue, pain complaints, or irritability, and they might have a negative impact on the quality of services provided. While searching for the reason for such high values of occupational burnout in a group of corporate employees and nurses, especially on the exhaustion subscale, it is vital to look especially into factors directly related to work.

However, the results showing high values on the depersonalization subscale in corporate employees were quite surprising (76.96% medium and high level *q*2 = 13.50) and did not coincide with the results of the study by Kowalska et al. (Me = 4, median) [16]. The current study in a group of nurses showed low scores on the depersonalization subscale, which may result from the relatively young group of respondents and their short period of employment, although a similar result was obtained in 2018 in our other study (Me = 6.0, median) [19].

We were unable to find studies presenting similar results with such significant differences in depersonalization values between the studied groups. The findings of Dobrowolska and Słazyk-Sobol may provide valuable insights into this observation. Their study compared the relationship between emotional labor among teachers and commercial service workers and the incidence of burnout. Commercial service workers, unlike teachers, exhibited a higher intensity of superficial interactions, such as pretending to display positive emotions while suppressing negative ones. These behaviors, often driven by the requirements of large companies demanding employees to consistently display positive emotions to customers, were associated with greater energy expenditure to control and adjust emotions. This, in turn, led to higher levels of exhaustion, reduced commitment, and emotional distancing from customers. Moreover, apart from distancing from customers, high scores on the depersonalization subscale could also indicate a pathological attitude toward one’s work environment, colleagues, and subordinates. Therefore, preventing the development of this component of burnout should be of particular concern to employers [17,20,21,22].

A reduced sense of personal accomplishment is the third dimension of occupational burnout. The analysis of our study revealed that low and medium levels of personal accomplishment were reported by 87% of corporate employees (*q*2 = 27.0) and 78% of nurses (*q*2 = 32.0). These findings align with those of Wilczek-Rużyczka, who reported low and medium levels of personal accomplishment in 96% of nurses surveyed, many of whom also experienced high emotional exhaustion and reduced personal accomplishment. A diminished sense of personal accomplishment can lead to a loss of meaning in one’s work, increased negative emotions, deteriorating health, and more frequent absences from work [23].

The relationship between gender and burnout incidence is complex. Previous studies have often shown that women are more susceptible to burnout due to their greater emotional involvement in work, reduced distance from professional responsibilities, and a stronger tendency to identify with others’ problems. This heightened involvement often results in higher levels of exhaustion compared to men. Conversely, men are more likely to adopt an instrumental approach in professional relationships, leading to higher depersonalization values. This instrumental attitude may explain the greater impersonal attitudes observed in men [23,24,25,26]. However, in our study, no significant gender-based differences in the incidence of occupational burnout were observed in either group.

A study conducted by Świątek, Milecka, and Uchmanowicz observed that single individuals tend to report higher rates of burnout compared to those in relationships. However, this finding was not confirmed by our study [26].

In contrast, our study revealed that nurses with children exhibited higher levels of emotional exhaustion. This may be attributed to additional responsibilities before and after work related to childcare or a lack of free time. Furthermore, nurses with children showed lower depersonalization values (*q*2 = 4.0 vs. *q*2 = 6.0). It can be hypothesized that individuals with children and families tend to be more caring, receive more support from their families, and experience a greater sense of purpose in their actions [10,25].

In the scientific literature, most studies suggest that higher earnings are generally associated with lower levels of burnout. For instance, a study by Kupcewicz and Szczypiński reported clear correlations between financial status and all three dimensions of occupational burnout. Financial status was found to significantly associated emotional exhaustion (*p* < 0.001), depersonalization (*p* < 0.04), and personal accomplishment (*p* < 0.001) [27]. In our study, financial status was significantly associated with emotional exhaustion only. Nurses with average earnings reported higher emotional exhaustion (*q*2 = 23.0), while those at the lowest and highest income levels reported lower exhaustion (*q*2 = 20.50 and *q*2 = 20.0). This may be due to the nature of roles held by nurses earning average salaries, which often involve both physical labor, requiring stamina and manual dexterity, and mental strain from working under time pressure and maintaining frequent contact with patients. However, no confirmation of this hypothesis was found in the available literature, and it should be regarded as speculative, requiring further investigation.

Despite the differing nature of work and tasks, similar results were observed among corporate employees. Employees earning the least and the most reported lower levels of emotional exhaustion (*q*2 = 37 and *q*2 = 27.50). In large corporations, those earning the least often perform tasks requiring lower competence and responsibility, typically at the start of their careers, and thus may not yet experience significant emotional exhaustion. Conversely, the highest earners often hold senior, responsible positions and may feel financially and professionally fulfilled, reducing their susceptibility to fatigue. In contrast, workers earning average wages often strive for promotions and higher salaries, investing significant energy into their work and frequently working overtime. This constant pursuit of advancement may lead to higher rates of emotional exhaustion.

Our study did not find a relationship between burnout levels and household size in either group, and, similarly, no references to this variable were identified in the literature.

In the nursing profession, improving competencies and obtaining higher professional qualifications are integral aspects of career development. These advancements provide greater independence and opportunities for promotion. In the group of nurses with higher education, lower values of exhaustion (*q*2 = 20 vs. average *q*2 = 23), higher levels of depersonalization (*q*2 = 5.0 vs. average *q*2 = 4.0), and lower job satisfaction (*q*2 = 32 vs. average *q*2 = 28) were observed. In contrast, the opposite was true for exhaustion and depersonalization in corporate employees, with those with higher education experiencing higher exhaustion (*q*2 = 32 vs. average *q*2 = 29) and lower depersonalization (*q*2 = 13 vs. average *q*2 = 18). Corporate employees with higher education similar to nurses reported lower job satisfaction (*q*2 = 28 vs. average *q*2 = 20). The fact that three-quarters of the respondents in this study had a university education may have contributed to the lack of a consistent protective effect of education. While continuous learning and competence enhancement are expected to boost self-esteem and self-confidence, this was not confirmed in our findings. Depersonalization, potentially manifesting as efforts to minimize contact with others, might also result in distancing oneself from those perceived as less competent. These relationships are likely more complex and resist simple interpretation [24,28,29].

Leisure plays a critical role in preventing occupational burnout. Increasingly, companies are implementing “corporate wellness” health programs aimed at improving employee well-being. A study by Parks and Steelman found that participation in such programs was associated with lower absenteeism and higher job satisfaction [27].

Our findings support the notion that the lack of holidays significantly contributes to greater emotional exhaustion and lower job satisfaction in both groups. This underscores the importance of recovery and relaxation in preventing burnout. Among nurses, only half reported taking holidays in the previous year. The reasons for this could include insufficient wages and a lack of flexibility in selecting holiday dates, as employees are often required to choose their vacation dates early in the year without the possibility of later adjustments. This inflexibility is frequently due to inadequate staffing levels in hospital wards.

In one study on occupational burnout among nurses working in inpatient hospices, female respondents highlighted the importance of more frequent opportunities for holidays as a key measure to counteract burnout [30]. Among the corporate employees surveyed, as many as 77% of the respondents went on holidays, whereas in the nurses’ group, only 50% of the respondents did, which may be due to the policy of large companies and corporations in which employees must take a mandatory holiday of no less than 14 consecutive calendar days.

The labor market reflects strong differentiation among age groups, leading to generational differences in attitudes toward work and careers. According to a study by Smolbik-Jęczmień from the University of Wroclaw, 1980 is a dividing line between two generational groups. Representatives of the over-50 generation are characterized by low occupational mobility and are proponents of a traditional career, most preferably in one company. For them, work is a value in itself, and changing it often raises the fear of losing the professional position achieved. The younger generation highly values work, which is the realization of passions and interests and enables development and balance between personal life and work. This younger generation is characterized by high professional mobility and openness to change. Our findings align with these generational characteristics. Among nurses, 58% had never changed jobs, which may be related to their average length of service (11.5 years) and the age distribution of respondents, with the oldest participant being 69 years old. Nurses who had never changed jobs were the most emotionally exhausted and exhibited the highest levels of depersonalization. In contrast, corporate employees had a shorter average length of service (7 years), and the oldest respondent was 50 years old. The frequency of job changes was much higher in this group: nearly 60% had changed jobs up to three times, and 30% had changed jobs more than three times. Only 11% had never changed jobs. These findings strongly align with Smolbik-Jęczmień generational distinctions [31]. The higher frequency of job changes among the younger generation is that the labor market is changing, and different career dimensions than those typical 30 years ago are beginning to appear. Nowadays, career development requires continuous investment in oneself, learning, and constantly gaining new experience or expanding one’s competencies (e.g., training, specialized courses, etc.) to become more competitive in the labor market, which is true, especially for corporate employees. Among nurses, the change in the labor market structure (transition to a worker’s market) over the past few years, which does not limit nurses to working only in one place and one country, may be important [32].

Research confirms that the nature and characteristics of work significantly influence the development of occupational burnout syndrome. Among the contributing factors, shift work plays a critical role. In a study by Amy Hoffman and Linda Scott, irregular working hours were identified as a predisposing factor for burnout syndrome. Nurses working an 8 h day shift reported lower stress levels compared to those working 12 h shifts [33]. In our study, most nurses worked in a two-shift system. Nurses working in a two-shift or three-shift system achieved higher levels of burnout, e.g., EE—*q*2 = 20.50 and 23.50 vs. 17.50; PA—*q*2 = 32 and 27, vs. 33, and especially significant on the DEP depersonalization subscale—*q*2 = 5.0 and 7.5 vs. 2.0. The lower job satisfaction of nurses working in a two-shift or three-shift system confirms, described in the literature, the negative impact of shift work on the quality of personal and professional life. Nurses often lack control over their shift schedules, which are typically imposed by supervisors. Shift work can result in decreased energy and increased fatigue, and night shifts specifically contribute to sleep problems, hinder recovery, and lead to faster energy depletion, irritability, impulsiveness, and a greater risk of workplace errors.

Among corporate employees, only 14% reported working under an irregular schedule. However, a significant correlation of irregular work systems was observed across all three subscales of burnout. These employees experienced higher emotional exhaustion, greater depersonalization, and lower personal accomplishment. Unlike nursing, little research exists on the relation of work systems on burnout among corporate employees, possibly because single-shift schedules predominate in most corporations [28,32,34,35,36].

Burnout syndrome is influenced not only by individual factors but also by working conditions and environments. Among the surveyed corporate employees, nearly 60% reported working overtime, which significantly contributed to higher levels of emotional exhaustion and depersonalization. In contrast, no statistically significant relationship was found between overtime and burnout among nurses, with only 19% of respondents reporting overtime work. However, the differences may also result from the authors’ incorrect formulation of the question or its misunderstanding by the respondents, which presumably may be due to different management of working time in the corporation and the hospital. In the case of corporate employees, the concept of “overtime” is clearly defined by the employer and understood by the employee—any extra time is, in most cases, paid for as an addition to the basic salary. In the case of nurses, the system works differently and usually involves receiving remuneration for extra hours in the form of a day off, which may have influenced a different understanding of the question [7,33]. Interestingly, corporate employees who worked overtime reported the highest levels of job satisfaction. Additionally, 85% of corporate employees indicated that their company had a bonus system, which included periodic bonuses, performance-based pay, or financial rewards. It can be speculated that spending longer hours at work improves performance, leading to additional compensation and, consequently, higher job satisfaction. However, confirming a relationship between job satisfaction and the bonus system would require further analysis [26].

Our research indicates that 41% of the nurses surveyed chose their profession out of a desire to help others, while for 25%, the choice of profession was determined by coincidence. Similar findings were reported in a 2017 study conducted among nurses working in Intensive Care Units. As previously described, occupational burnout is more likely to affect ambitious individuals who begin their careers with enthusiasm and a sense of mission. However, some scientific publications partially contradict this claim. For instance, Sobczak’s research on professional motives versus occupational burnout found that respondents who chose the nursing profession out of a desire to help others exhibited lower levels of burnout and depersonalization and higher levels of personal accomplishment. In the group of corporate employees, there was no clear answer as to the factors that determined their choice of profession, which may be caused by the diversity of the group, their various education, and the different departments in which they worked. It can be found in the literature that a low level of motivation or a lack of understanding of the motive of one’s job can exacerbate job dissatisfaction, and frustration, which in turn can exacerbate occupational burnout. To confirm or disprove the above statements from our research, additional analysis of the relation of the factors influencing one’s career choices on the various subscales of occupational burnout would need to be conducted in the future [7,33,37]. In the study group, 70% of nurses expressed satisfaction with their work, and 66% stated they would not want to change their profession. Similar findings were reported by Grzywna and Cieślik, where 63% of nurses declared job satisfaction [38], and by Majchrowska and Tomkiewicz, where over 75% emphasized that they would not change their profession [39]. Additionally, Kędra and Sanak’s study found that 86% of respondents viewed their work as important and necessary [40]. Our findings align with these results, confirming high levels of job satisfaction among nurses. However, when interpreting these findings, it is essential to consider the employment duration of the surveyed nurses. Some had been in the profession for as little as 2 months. For individuals with such short periods of employment, it is challenging to discuss occupational burnout or the desire to change jobs as their experiences in the profession are still limited and may not fully reflect long-term challenges.

Among corporate employees, the majority (63%) reported job satisfaction. Similar findings were observed in a study by Bakonyi and Bilnik, where 64% of senior and lower-level employees expressed general satisfaction with their jobs. However, an interesting phenomenon emerged: despite most corporate employees expressing job satisfaction, nearly the same percentage (59%) indicated a desire to change their jobs [41].

## 6. Conclusions

This study indicates that most corporate employees and nurses surveyed experienced occupational burnout.

The analysis revealed higher burnout levels on the exhaustion and depersonalization subscales among corporate employees. Both groups demonstrated a reduced sense of personal accomplishment, but nurses exhibited a higher sense of personal accomplishment and expressed less desire to change their profession. Corporate employees living in urban areas were less emotionally exhausted but also less satisfied with their jobs compared to those in rural areas. Among corporate employees, those rating their financial status as good reported the least emotional exhaustion, lowest levels of depersonalization, and least job satisfaction. Nurses who had never changed jobs were more cynical but reported higher personal accomplishment values. Both corporate employees and nurses who did not take time off were more emotionally exhausted and experienced a reduced sense of personal accomplishment. Corporate employees working overtime were more tired and cynical but paradoxically reported a greater sense of personal accomplishment.

### 6.1. Practical and Theoretical Implications

These findings suggest that managers and policymakers should prioritize improving working conditions for both nurses and corporate employees by implementing strategies such as enhancing access to information, ensuring fair resource distribution, and providing professional development opportunities. Preventive actions against burnout should be implemented at three levels: organizational, individual–organizational, and individual. At the organizational level, measures include recruiting new employees with predispositions for specific roles, providing training for skill acquisition and professional knowledge (including stress management), improving physical and psychosocial working conditions, enhancing communication and collaboration, increasing job control, ensuring the effective utilization of skills, managing workloads, and promoting workplace safety. At the individual–organizational level, actions include fostering support from superiors and colleagues, aligning individual job opportunities with requirements, clarifying roles and responsibilities, and adopting a participatory management model. At the individual level, strategies involve developing relaxation techniques for managing tension, self-improvement, recognizing and regulating mental states, reducing irrational thinking through cognitive processes, accepting and modifying unpleasant experiences, and maintaining control over them. Other measures include engaging in physical activity, adopting a healthy diet and lifestyle, and developing skills such as delegation, negotiation, goal-setting, and problem-solving. Consulting specialists for professional or personal challenges are also encouraged.

Implementing these approaches is expected to reduce burnout among nurses and corporate employees.

### 6.2. Limitations of the Study

The study has several limitations. One notable limitation is the age of the respondents, suggesting the need to repeat the study with more comparable groups to confirm the findings. Consequently, the results should be interpreted with caution. At the same time, our study is the first attempt in Poland to describe the universal factors that can influence occupational burnout both in corporations and healthcare units.

The study results should be interpreted with caution due to the limitations of cross-sectional studies (e.g., limited generalizability, random bias, and recall error) and limitations resulting from the sampling method. It is also possible that differences in burnout are due to the nature of the work performed or other factors not identified by us.

Future studies should be conducted, in separate groups of employees, to clarify doubts, among others, regarding the low work efficiency of Polish employees and the reason for burnout on a representative sample, enabling the generalization of the results and their practical implications, as well as dedicated preventive measures.

This research task was financed within the framework of the grant of the Ministry of Science and Higher Education (MNiSW) for maintaining the research potential of the Faculties of the Jagiellonian University CM (K/ZDS/004675).

## Figures and Tables

**Table 1 healthcare-13-00123-t001:** Socio-demographic variables of nurses and corporate employees participating in the survey.

Socio-Demographic Variables										
	Nurses					Corporate Employees				
	n	x	%	*q*2	*q*1–*q*3	n	x	%	*q*2	*q*1–*q*3
Gender										
Female	123		97.62			146		71.57		
Male	3		2.38			58		28.43		
Age	123	37		37.03	26–50	204	29		28.80	26.70–33.57
Education										
Higher	93		73.81			179		87.75		
Secondary	33		26.16			25		12.25		
Place of residence										
City	84		66.67			194		95.10		
Country	42		33.33			10		4.90		
Marital status										
In a relation	98		77.78			136		66.67		
Single	28		22.22			68		33.33		
Having children										
Yes	63		50.00			42		20.59		
No	63		50.00			162		79.41		
Household size										
One person	41		32.54			65		31.86		
2–5 people	77		61.11			137		67.16		
More than 5 people	8		6.35			2		0.98		

n—number of observations; x—average arithmetic; %—percentage; *q*1, *q*2, and *q*3—quartile value.

**Table 2 healthcare-13-00123-t002:** Employment history and financial status of nurses and corporate employees participating in the study.

	Nurses				Corporate Employees			
	n	%	*q*2	*q*1–*q*3	n	%	*q*2	*q*1–*q*3
Length of employment	126		11.50	1.5–25	204		7.00	4–10
Work system								
Single-shift	32	25.40			176	86.27		
Two-shift	78	61.90			17	8.33		
Three-shift	16	12.70			11	5.39		
Bonus system								
Yes	0	0			204	100		
No	126	100			0	0		
Working at weekends, public holidays, and night shifts								
Yes	118	93.70			123	60.30		
No	8	6.30			81	39.70		
Extra work								
Yes	31	24.60			29	14.22		
No	95	75.40			175	85.78		
Frequency of workplace changes								
Never	73	58.00			23	11.00		
1–3 times	47	37.00			120	59.00		
More than 3 times	6	5.00			61	30.00		
Self-assessment of financial status								
Very good and good	51	40.00			134	66.00		
Average	64	51.00			61	30.00		
Bad	11	9.00			9	4.00		
Going on holidays in the last year								
Yes	63	50.00			157	77.00		
No	63	50.00			47	23.00		

n—number of observations; %—percentage; *q*1, *q*2, and *q*3—quartile value.

**Table 3 healthcare-13-00123-t003:** Prevalence of occupational burnout in the studied group of nurses and corporate employees.

Occupational Burnout According to Maslach	Nurses				Corporate Employees			
	n	%	*q*2	*q*1–*q*3	n	%	*q*2	*q*1–*q*3
Emotional Exhaustion			20.5	14.0–27.75			31.5	16.0–41.0
Low	42	33.33			54	26.47		
Medium and high	84	66.67			150	73.53		
Depersonalization			5	2.0–8.0			13.50	7.0–19.25
Low	83	65.87			47	23.04		
Medium and high	43	34.13			157	76.96		
Personal accomplishment			32.0	25.0–38.0			27.00	19.75–34.0
Low and medium	99	78.57			179	87.75		
High	27	21.43			25	12.25		

n—number of observations; %—percentage; *q*1, *q*2, and *q*3—quartile value.

**Table 4 healthcare-13-00123-t004:** Comparison prevalence of burnout in nurses and corporate employees according to socio-demographic variables.

Variables		Nurses						Corporate Employees					
		EX		DEP		PA		EX		DEP		PA	
		n	*q*2 (*q*1, *q*3)	n	*q*2 (*q*1, *q*3)	n	*q*2 (*q*1, *q*3)	n	*q*2 (*q*1, *q*3)	n	*q*2 (*q*1, *q*3)	n	*q*2 (*q*1, *q*3)
Education	Higher	93	20 (14.0; 27.0)	93	5.0 (2.0; 10.0) *	93	32.0 (26.0; 38.0)	179	32.0 (16.0; 41.0)	179	13.0 (7.0; 20.0)	179	28.0 (21.0; 34.0)
	Secondary	33	23.0 (14.0; 26.0)	33	4.0 (2.0; 6.0) *	33	28.0 (24.0; 38.0)	25	29.0 (19.0; 42.0)	25	18.0 (9.0; 19.0)	25	20.0 (14.0; 34.0)
Having children	Yes	63	23.0 (16.0; 30.0) *	63	4.0 (1.50; 6.0) *	63	30.0 (24.0; 36.50)	42	37.0 (19.0; 45.75)	42	14.50 (7.25; 21.75)	42	27.50 (18.25; 33.75)
	No	63	18.0 (13.50; 25.50) *	63	6.0 (3.0; 10.0) *	63	33.0 (26.50; 38.0)	162	31.0 (16.0; 40.75)	162	13.0 (7.0; 19.0)	162	27.0 (20.0; 35.0)
Place of residence	City	84	20.0 (13.0; 28.0)	84	4.50 (2.0; 7.25)	84	33.0 (24.75; 38.0)	194	31.0 (16.0; 41.0) **	194	13.0 (7.0; 19.0)	194	28.0 (21.0; 35.0) ***
	Country	42	23.0 (16.25; 27.0)	42	5.0 (3.0; 9.75)	42	29.50 (25.25; 35.75)	10	40.5 (37.50; 51.0) **	10	18.50 (13.25; 21.25)	10	13.50 (10.0; 17.75) ***
Work system	One-shift	32	17.50 (11.75; 24.0)	32	2.0 (0.75; 6.0) **	32	33.0 (27.75; 38.25) *	176	31.0 (16.0; 41.0) *	176	12.50 (6.0; 19.0) **	176	28.0 (20.75; 35.0) *
	Two-shift	78	20.50 (14.25; 28.75)	78	5.0 (2.25; 8.75) **	78	32.0 (25.25; 37.75) *	17	37.0 (33.0; 46.0) *	17	19.0 (17.0; 21.0) **	17	23.0 (18.0; 31.0) *
	Three-shift	16	23.50 (15.50.39.0)	16	7.5 (4.0; 14.25) **	16	27.0 (17.0; 37.50) *	11	19. 0(9.0; 22.0) **	11	19.0 (9.0; 22.0) **	11	20.0 (14.50; 27.50) *
Frequency of workplace changes	Never	73	21.0 (14.0; 28.0)	73	5.0 (3.0; 10.0) *	73	33.0 (26.0; 39.0) ***	22	30.0 (17.75; 38.0)	22	13.0 (8.25; 18.75)	22	24.50 (22.0; 31.0)
	1–3 times	47	20.0 (16.0; 27.0)	47	4.0 (1.50; 6.0) *	47	32.0 (24.0; 36.50) ***	120	32.0 (16.0; 42.0)	120	13.0 (6.75; 19.25)	120	36.50 (18.0; 34.0)
	>3 times	6	21.0 (14.75; 28.0)	6	3.50 (2.25; 4.75) *	6	30.0 (27.0; 30.75) ***	62	31.50 (15.25; 40.75)	62	16.0 (7.25; 19.75)	62	30.50 (21.25; 36.0)
Self-assessment of financial status	Very good and good	51	20.0 (15.50; 28.50)	51	6.0 (3.50; 9.50)	51	31.0 (23.50; 37.50)	134	27.50 (15.0; 39.75) ***	134	11.0 (5.0; 18.0) ***	134	28.0 (20.25; 35.0) *
	Average	64	20.50 (13.75; 27.0)	64	4.0 (2.0; 7.0)	64	32.0 (26.75; 39.0)	61	37.0 (24.0; 45.0) ***	61	18.0 (9.0; 22.0) ***	61	27.0 (19.0; 34.0) *
	Bad	11	23.0 (17.0; 27.50)	11	3.0 (1.50; 5.50)	11	33.0 (24.50; 36.0)	9	40.0 (37.0; 45.0) ***	9	23.0 (14.0; 26.0) ***	9	21.0 (19.0:27.0) *
Going on holidays in the last year	Yes	54	19.0 (14.0; 25.0) *	54	5.0 (2.25; 9.0)	54	33.0 (26.25; 38.0) *	157	30.0 (15.0; 40.0) *	157	13.0 (7.0; 19.0)	157	29.0 (22.0; 35.0) ***
	No	54	23.0 (15.25; 30.0) *	54	4.0 (2.0; 8.0)	54	30.0 (24.0; 36.0) *	47	39.0 (20.0; 47.50) *	47	16.0 (8.0; 21.0)	47	21.0 (16.0; 31.0) ***
Job satisfaction	Yes	88	19.0 (14.0; 27.0) *	88	5.0 (2.0; 8.0)	88	33.0 (27.0; 38.0)	128	22.50 (12.0; 33.25)* *	128	10.0 (4.75; 18.0) ***	128	31.0 (23.0; 37.0) *
	No	38	23.50 (16.25; 29.75) *	38	5.0 (2.0; 8.75)	38	28.50 (22.25; 35.25)	76	42.0 (36.75; 47.25) **	76	18.0 (11.75; 23.0) ***	76	21.0 (16.75; 27.25) *
Willingness to change jobs	Yes	43	24.0 (17.0; 33.50) **	43	5.0 (1.50; 8.0)	43	39.0 (21.50; 35.50)	120	38.50 (29.0; 46.0) ***	120	17.0 (9.0; 22.0) ***	120	24.0 (18.0; 31.0) ***
	No	83	18.0 (13.50; 25.50) **	83	5.0 (2.0; 8.0)	83	33.0 (26.50; 38.0)	84	19.0 (11.0; 29.50) ***	84	8.50 (3.0; 17.0) ***	84	33.0 (23.75; 38.0) ***

n—number of observations; *q*1, *q*2, and *q*3—quartile value, statistically significant: *p* < 0.05 *; *p* < 0.01 **; *p* < 0.001 ***. EX—emotional exhaustion, DEP—depersonalization, and PA—personal accomplishment.

## Data Availability

Data are contained within the article.

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
