# Peer review of "Occupational Burnout in Nurses and Corporate Employees in Małopolska Region, Poland"

_healthcare, 2025, doi:10.3390/healthcare13020123_

Round 1
Reviewer 1 Report (Previous Reviewer 2)
Comments and Suggestions for Authors
Dear Authors,
Thank you very much for allowing me to review your research again. I enjoyed the reading. The changes introduced significantly improve the article. However, I propose different changes, e.g., introduction, practical and theoretical implications, among others. I especially liked the results section as the analysis is very deep and provides significant data. The different relationships they establish generate value to the research. The discussion of the results has also improved substantially. Adequately comparing your results with other previous results is essential to give your work the value it deserves.
Abstract
Better Abstract based on all suggestions.
1. Introduction
Suggestion: The authors should do a better job of establishing the relationship between Burnout, extended working hours, low wages and low productivity. Currently, this relationship is not clear in specific contexts, such as Poland.
It is important to better relate stress, chronic work stress and Burnout. In fact, stress is not necessarily related to Burnout. It is important for the authors to define stress, chronic job stress and Burnout. The authors explain the dimensions of Burnout according to Christina Maslach but do not adequately define the concept.
Being able to compare factors that induce Burnout in different work groups is very interesting. The authors state that there are no similar studies that analyze Burnout in corporate employees and active nurses simultaneously. However, are there studies that analyze Burnout among corporate employees and nurses separately? It is essential to better define the originality of this research, what does their study contribute to the existing literature? Finally, the authors should better define the general objective of this study and the specific objectives. It is important that the authors delve more deeply into the concept of Burnout and its origins in the population groups studied. Since there is no theoretical framework, the introduction is conceptually limited.
3. Materials and Methods
Suggestion: Include the overall AVE value of the MBI-HSS questionnaire and the separate AVE values for each dimension. How did the authors address common method bias?
It is important to better explain the type of sampling used.
4. Results
Suggestion: It is important that the authors take into account that the sample has a gender bias, that is, that it is made up almost exclusively of women. They may consider this circumstance as a potential limitation of the study.
Authors should keep in mind that depersonalization and cynicism are not synonymous. It is important to unify concepts.
5. Discussion
Suggestion: The higher emotional exhaustion of corporate employees with respect to nurses, why don't you compare it with previous studies?
The justification about low levels of depersonalization seems to me to be accurate. A young population with short work experience is less exposed to the depersonalization that tends to manifest when the imbalance between resources and demands is sustained over time.
It is important to include a separate section on practical and theoretical implications. Such a thorough study needs to incorporate useful practical and theoretical implications.
6. Conclusion
Suggestion: The authors have a research question: what accounts for the low productivity of Polish workers? and at least one general objective. The conclusions are an excellent opportunity to give a brief answer to the research question and the different objectives. The conclusion section is not optimal for including practical implications.
Limitations of the study:
Suggestion: Include future research. Expand limitations, e.g., gender of respondent.
Author Response
Please see attachment

Reviewer 2 Report (New Reviewer)
Comments and Suggestions for Authors
This study investigated the correlations between sociodemographic and occupational factors and the sub-dimensions of burnout. However, the aim and analyses of the study, as well as its key findings, were unclear and required reconsideration. Specifically, it was ambiguous whether the authors aimed to examine disparities in burnout between the two occupational groups or to identify factors associated with burnout syndrome within each group. Below were my comments on the manuscript:
The stated aim of this study was to compare burnout levels between nurses and corporate employees. However, the analyses were conducted separately for each occupational group, and no statistical comparison between the two groups was performed. Two critical aspects needed to be considered:
- As shown in Table 1, there were significant differences between the groups in terms of gender, age, residential region, and family composition.
- The sampling procedure did not fully ensure the representativeness of the two groups.
These factors can raise substantial concerns regarding the validity of directly comparing burnout scores between the two occupaitonal groups.
Causal Language
As this study was based on a cross-sectional design, causal language—such as "effect," "impact," or "affect"—needed to be replaced with terms that reflected associations or correlations throughout the manuscript.
Study Aim
Although the stated objective of this study was to "compare the prevalence and intensity of burnout syndrome in two occupational groups," the analyses descriptively examined correlations between sociodemographic and occupational factors and burnout syndrome within each group. Statistical comparisons of burnout syndrome between the two occupational groups were not conducted, yet comparisons were inappropriately discussed in the text.
Methods
Why were regression analyses not performed? Regression analysis may be useful not only for exploring factors associated with burnout within each occupational group but also for statistically comparing burnout differences between the groups.
Tables
- Table 1: The presentation is unclear. For instance, what does "x" indicate? The table is needed to be reformatted for clarity and consistency with other tables. Additionally, it seems feasible to merge Tables 1 and 2 for a more streamlined presentation.
- Table 3: The term "incidence" is inappropriate and needed to be replaced with "prevalence" throughout the manuscript. Also, clarification is needed regarding what Q2 and Q1–Q3 represent.
- Table 4: P-values should be presented as exact three-digit values rather than using asterisks. Also, due to the table's large size, dividing it by occupational group may improve readability.
Discussion
The discussion is overly lengthy and, I believe, fails to effectively convey the study's key findings. While the introduction highlights the associations between occupational factors and burnout, the discussion provides descriptions of all factors associated with burnout, making the manuscript overly broad and less focused. The authors are encouraged to prioritize less-known factors, particularly occupational factors that can inform interventions, rather than addressing all associated factors.
Specific points:
- Line 296: The phrase "high emotional exhaustion" is inaccurate and should be revised to reflect "high or medium levels".
- Lines 296–301: Statistical comparisons between the two occupational groups were not conducted; however, comparisons were inappropriately made in the discussion.
- Line 309: Clarification is needed for the meaning of "Me=4." Does it refer to the median? The same clarification is needed for "Me=6.0" in line 312.
Limitations
The limitations section should be expanded to include the constraints of the cross-sectional design, recall bias, and the limited generalizability of the sample.
Round 2
Reviewer 1 Report (Previous Reviewer 2)
Comments and Suggestions for Authors
Dear Authors,
Thank you very much for allowing me to review your article again. I have enjoyed reading this new version and the small modifications you have included have increased the final quality of the document.
I agree with the authors, the relationship between Burnout syndrome and low productivity would need an independent study. The authors adequately explain the relationship between chronic stress and management-related stress with respect to Burnout syndrome. I agree with the authors, the scales proposed by Maslach have severe limitations. Using categorized scores is an appropriate option. The authors justify the gender bias as in Poland only 2% of nursing professionals are men. Finally, the authors include a separate section on practical implications, lines 546-568 and improve their limitations and future lines of research, lines 576-583. Overall, I congratulate them on their article.
Author Response
Dear Sir,
Thank you very much for your friendly review of our article. Your advices were very helpful in improving its quality.
Sincerely
Authors
Reviewer 2 Report (New Reviewer)
Comments and Suggestions for Authors
Thank you for addressing my comments.
Although my concerns regarding the methodological aspects have not been fully addressed, I agree with the authors' appeal of the necessity of this research. However, my primary concern, particularly the differing characteristics of the two groups and the possibility that each group may not be representative of their respective occupations, must be thoroughly discussed as a limitation. This issue requires careful consideration when making comparisons.
Thank you.
Author Response
Please see the attachment

This manuscript is a resubmission of an earlier submission. The following is a list of the peer review reports and author responses from that submission.
Round 1
Reviewer 1 Report
Comments and Suggestions for Authors
The study topic is interesting, but this study has several methodological weaknesses.
First of all this study requires moderate language revisions.
Second, the study title should be more precise and define the study population and settings (e.g., in the Małopolskie region, Poland)
Third, the Authors did not provide correct citations, e.g., lines 38-46, 79-90 in the introduction section. This repeats both in the introduction and discussion.
Sentences like "Work is an important part of human life" should be removed because this is not scientific wording and this sentence does not contribute to this scientific paper.
The most important comment is related to the study sample:
- there is a lack of information on the inclusion/exclusion criteria
- sampling methods are unclear
- there is a lack of data on the response rate
- the authors stated "standard questionnaire" but in fact, this is not true as there is a lack of a standard set of questions on demographic characteristics
- the authors would like to compare burnout between nurses and corporate employees but there is a lack of data on how nurses and corporate employees were selected
- there is a huge risk of bias in this study resulting from the sampling and recruitment of the participants
The discussion is too extensive and should be more precise
There are numerous limitations related to the study design that were not metioned by the Authors
Conclusions should be revised as a well-organized text rather than numerical points.
Comments on the Quality of English LanguageThis study requires extensive language revisions. The Authors used inappropriate words (e.g., line 28, line 79 etc.).
Author Response
Dear Reviewer:
Thank you very much for your comments on our work. We have completed the text and made corrections according to the instructions and additionally
- We have supplemented our text with:
Inclusion and exclusion criteria, we added the methods of group selection, added a survey attendance index, ensnared the title of the paper and changed the introduction.
- The study group was joined by volunteers who were invited to the study by the authors. The authors were not employees of any of the institutions participating in the study and did not gain any benefit from the publication of the data of the above study.
- Fixed errors resulting from incorrect translation of the term personal accomplishment.
- The impact of burnout on management was not studied due to the low size of the group.
Yours sincerely,
Authors
Reviewer 2 Report
Comments and Suggestions for Authors
Occupational Burnout in Nurses and Corporate Employees
Abstract
It is generally good
1. Introduction
Dear authors,
I like the initial part of the introduction because you are already approaching the country under study and its characteristics. Many introductions tend to be very general and avoid delimiting a specific problem.
However, the low productivity and profitability of Poles needs further explanation. That is, why are Poles not productive, what studies prove this assertion. Also, how much does the average worker in Spain, Portugal or Greece earn? The comparison with Norway, Luxembourg and Ireland is possibly not the best.
Undoubtedly, stress caused by excessive dedication to work can paradoxically lead to poorer work performance.
The three dimensions of burnout (Christina Maslach) are well explained. Table 1 is good, but I would have liked the factors associated with burnout to be explained in the text. Tables in general are not used too much as a reference element.
The direct effect of technology on depersonalisation requires reference author(s).
The persons analysed are well delimited and the research question is well formulated.
The introduction is a great opportunity for the authors to specify what their study contributes in relation to previous studies. In other words, what knowledge gap they intend to fill. Moreover, are there other similar studies in Poland? It is important to differentiate yourself.
2. Aim of the Study
This part is well specified.
3. Material and Methods
Why the Malopolska Region, any special reason for choosing such a specific area?
Having the endorsement of the Bioethics Committee is important.
In addition to informed consent the participant had the option of voluntary withdrawal, data protection and so on.
They did not apply any exclusion criteria, e.g. minimum years of work. It is easier for a person who has worked for years in the same job to experience burnout than a newcomer.
It would be interesting to explain the type of sampling applied.
By including in this section, the instrument used I consider it necessary to present the AVE values for each dimension.
A burnout scale translated into Polish was used. How was the translation done?
Statistical analysis is fine. Many researchers usually omit the non-normal distribution of the data.
The ANOVA test is fine as is the statistical package used.
4. Results
4.1. Demographic and Work Condition Variables
The description of the participants is good. Table 2 is fine.
The description of the shifts seems to me to be accurate and necessary. It is also important to specify the percentages of participants who used to work overtime. Table 3 is good.
4.2. The Incidence of Occupational Burnout
The percentages of emotional exhaustion and the third dimension of burnout are very worrying. Table 4 is well specified.
The difference in depersonalisation rates (nurses in higher education versus nurses in secondary education) does not seem to me to be so significant. Also, are they nurses and nursing assistants?
The rest of the results are well described and compared between the two study groups. Table 5 is fine.
5. Discussion
Remember to introduce your research question. The results and your discussion will determine an answer to your initial question. Beware of one thing, the third dimension of burnout is self-realization. In your case you use job satisfaction as a synonym. It is not exactly the same thing. Evaluate this suggestion.
The initial comparison between corporate employees and nurses is fine. However, they repeat the results. The most important thing is not the description of the results but to go deeper into them.
The high levels of depersonalisation among corporate employees, they do not discuss them? why do they not match the results of Kowalska et al. (2005)? are there no more recent studies?
The Wilczek-Rużyczka study is from what year? Could you please compare your results with the results of this publication in more depth? This author is known for her analysis of the burnout phenomenon.
Reference 32 is missing.
The discussion includes descriptions from a theoretical framework. However, it is fine.
The susceptibility to emotional exhaustion among women has been analysed in depth. There are also studies that show the opposite and others that show a higher level of depersonalisation in men (for example your study).
The data on single people is very interesting.
The explanation for the higher emotional exhaustion in nurses with children is right. So are the levels of depersonalisation.
The relationship between burnout and financial status is very interesting. Contrary to your results there are many studies that positively relate emotional exhaustion to highly paid positions (management).
It would be interesting to further discuss the results of emotional exhaustion, depersonalisation and self-realization among nurses with more or less schooling.
Remember to review the term job satisfaction.
Reviewing the influence of rest is very good. Also, the differences between groups.
The generational study is very good. It is very common that people with very long working careers suffer higher levels of emotional exhaustion.
The analysis of shifts and their effect on burnout is very good. The same goes for overtime.
Professional choice is key to reducing stress and increasing the feeling of personal self-realization.
6. Conclusions
The conclusions are an opportunity to answer the research question. I suggest eliminating the numbers and giving the conclusions more continuity. Their study and the depth of their discussion allows for very important conclusions. Work needs to be done on that part.
7. Limitations of the study:
The limitations can be improved, why didn't they include a section on practical implications?
Author Response
Dear Reviewer,
Thank you very much for your comments on our work.
- The term profitability was added to our text accidentally, it was a mistake and it has already been removed
- We have decided to remove table 1 and describe the factors as you suggested in the text
- We have not found any literature that would refer directly to the influence of technology on depersonalization.
- We appreciate the suggestion regarding the originality of the work, we have added an appropriate text.
- The choice of the study region and the study group was dictated by the availability of research
- We supplemented it with a cloister in which we wrote about the voluntary participation, informed consent and the possibility of resignation from the test
- We have also supplemented the text with an exclusion criterion.
- The study group of nurses and corporate employees voluntarily agreed to participate in the study after being invited. We invited everyone who met the criteria and worked in the companies that wanted to cooperate with us to participate in the survey.
- We have supplemented the text with the missing values of the mean article for continuous variables. In the case of data on the three dimensions of burnout according to Maslach, the author of the scale recommends not using average values.
- The MBI questionnaire is an original text, translated into Polish by the Maslach research team and purchased by us in this form for our study.
- There were no nursing assistants in the study group of nurses, because there is no such profession in the Polish health care system.
Yours sincerely, the authors
Round 2
Reviewer 2 Report
Comments and Suggestions for Authors
Dear authors,
I would like to help you improve your work. However, I need a letter specifying the changes you have made to your article based on my suggestions. I have not seen such a letter.
Secondly, I need a finished article that is easy to read, with all corrections in a different colour. The article I have received I cannot evaluate properly.
Your work and effort are significant, but I need something more from the authors.
I hope they understand my position, as I owe it to Healthcare and the quality standards of the journal.
Kind regards
